# Resource Utilization of Lake Sediment to Prepare "Sponge" Light Aggregate: Pore Structure and Water Retention Mechanism Study

Yu Huang [1,2,3], Kunpeng Li [1], Chi Zhou [4], Xiaotian Du [1], Jiangnan Peng [1], Baowen Liang [1], Ziyi Ding [1] and Wen Xiong [1,2,3,*]

1   School of Civil Engineering, Architecture and Enivironment, Hubei University of Technology, Wuhan 430068, China
2   National Engineering Research Center for Advanced Technology and Equipment of Water Environmental Pollution Monitoring, Wuhan 430068, China
3   Hubei Key Laboratory of Ecological Restoration for River-Lakes and Algal Utilization, Wuhan 430068, China
4   Hubei Water Resources Research Institute, Hubei Water Resources and Hydropower Science and Technology Promotion Center, Wuhan 430070, China
*   Correspondence: 20171017@hbut.edu.cn; Tel.: +86-13907164835

**Abstract:** Nitrogen, phosphorus, and metals' pollutants discharged from industrial sources eventually accumulate in lake sediment, hence increasing the difficulty of sediment treatment and disposal. In this work, the water storage ceramsite is prepared from dredged lake sediment and cyano-bacterial powder. The effects of pyrolysis temperature and cyanobacterial sediment on the porosity of ceramsite were investigated. The results showed that the pyrolysis of organic matter and the de-composition of compounds or salts can produce gas, causing a rich pore structure inside the ceramsite. When the temperature increased to 1150 °C, vitrification would collapse the pore structure inside the material. At the cyanobacterial-to-sediment ratio of 3:7, the porosity and water absorption of the material could reach 81.82% and 92.45% when the pyrolysis temperature was 500 and 1050 °C, respectively. The internal macropore structure of ceramsite improved the water absorption performance, and the mesoporous structure was responsible for its long water release time and stable water release structure. The ceramsite exhibited a superior metals' retention effect. Under different pH and temperature conditions, the consolidation rates of Fe, Ni, Mn, Cr, and Pb in ceramsite were all more than 99%, suggesting the safety of the material in environmental applications. This study demonstrates the feasibility of the resourceful production of water storage ceramsite from lake sediment and cyanobacterial slurry, which helps to reduce the impact of solid waste on the environment. Thus, this work provides a practical basis for guiding water storage ceramsite in the construction of sponge cities.

**Keywords:** water storage materials; cyanobacterial; pyrolysis; sediment resources; sponge city

## 1. Introduction

The rapid development of industrialization is accompanied by the discharge of large amounts of wastewater and exhaust gases. Industrial production discharges pollutants' access to natural water bodies through the tailwater discharge of sewage treatment plants, surface runoff, and atmospheric deposition, which eventually sinks into sediment [1]. The endogenous release of lake sediment leads to the increase of nutrient concentrations in water. Nutrients such as nitrogen and phosphorus will trigger the explosive growth of cyanobacteria in water, the decrease of dissolved oxygen in water, the deterioration of water quality, the degradation of lake functions, and the eutrophication of the water body. Eutrophic lakes cause damage to the utilization of water resources and seriously affect the operation of water plants. Eutrophic water will corrode the pipe network and deteriorate the water quality of the effluent, which is not conducive to production and life [2]. Metals

in sediments are toxic to submerged plants and aquatic animals. Some metals can also be enriched through the food chain, ultimately causing harm to human health.

The treatment of eutrophic lakes adds phosphorus-locking agents and other reagents to make in situ solidification, physical dredging, and submerged plant regulation to control endogenous pollution. Although the physical dredging method can directly remove the cyanobacterial floating on the water surface and the sediment on the water bottom, the secondary pollution of a large number of solid wastes such as sediment and cyanobacterial slurry generated after dredging has become a new environmental problem [3–5]. These solid wastes have the characteristics of a large amount of organic content, complex metals' composition, high water content, and difficult dehydration, causing great difficulty for treatment and disposal [6]. At present, landfills and incineration are commonly used for treatment and disposal. Sediment landfills not only take up a lot of land resources but also pollute soil and groundwater (leachate) because of the metals in sediment and cyanobacterial slurry. Organic matter also continues to produce greenhouse gases such as methane during the landfill process [7]. Incineration is better for sediment and cyanobacterial slurry reduction with high organic matter content. However, it costs intense energy, and emissions of greenhouse gases such as $CO_2$ are an important reason for aggravating the change of the global greenhouse effect [8,9].

Therefore, solid waste disposal, such as substrate and cyanobacterial slurry, using resource-based means has received attention from researchers [10–12]. Riley et al., proposed the preparation of ceramsite using non-clay materials, discovered the chemical composition required for the preparation of ceramsite, and proposed the well-known ternary phase diagram [13]. The component analysis revealed that the composition of the subsoil, such as $SiO_2$ and $Al_2O_3$, was within the range of conditions required for the firing of ceramsite. This makes it a feasible way to realize sediment recycling by heat treatment [14]. With the promotion of sponge city construction, the Chinese government requires cities to be like sponges in adapting to environmental changes and responding to natural disasters and to show other aspects of good "resilience". The construction of sponge cities can improve the function of urban ecosystems. The research and development of "sponge city" construction materials that can absorb, store, percolate, and purify water has recently received great attention. Huang et al., used the sediment of Bei'an River in Mudanjiang City as raw material and verified the possibility of preparing ceramsite from sediment by designing orthogonal experiments to arrive at the optimal conditions for the preparation of ceramsite [15]. Li et al., prepared a ceramsite filter material from lake sediment, fly ash, and sewage sludge and analyzed the relationship between burn loss rate, swelling rate, and sintering temperature, aiming at the utilization of solid waste such as lake ceramsite [16]. Li et al., used iron tailings and fly ash as raw materials to prepare a porous ceramsite filter material, which saved resources and achieved the purpose of "treating waste with waste" [17]. Mi et al., used red mud, fly ash, and bentonite as raw materials to prepare high-strength ceramsite, which improved the strength performance of ceramsite [18]. Due to the poor water-absorption performance of ceramsite prepared from sediment alone, it cannot satisfy porous materials for sponge city construction. Therefore, researchers have also conducted a lot of research on pore-forming agents. Cai et al., prepared high-efficiency water-absorbing ceramsite with a water absorption of 66.71% from dredged lake sediment and fly ash [19]. Zhang et al., explored the effect of biomass addition on the water-absorption performance of fly ash ceramsite [20]. Zhong et al., prepared porous ceramsite using electric porcelain waste slag as raw material and starch as a porogenic agent to research the trends of the water-absorption effect and the compressive strength of ceramic pellets [21]. Inspired by the biomass porous agent, Zhao et al., creatively used cyanobacterial slurry as a porous agent in the preparation of porous light aggregate using lake sediment. Cyanobacterial slurry has a high organic matter content and high calorific value, which can not only be used as a porous agent when mixed with sediment for heat treatment but also can reduce energy consumption during heat treatment and can reduce costs [22].

The preparation of porous materials from lake sediment is in its infancy and the research is mostly focused on the water-absorption performance of the materials, how-

ever, the research on the water absorption mechanism is not deep and the research work on the water release mechanism of the materials has not yet been started. This greatly restricts the application of porous materials prepared from sediment resources in sponge city construction.

This paper's research project is porous ceramsite prepared from dredged lake sediment and cyanobacterial slurry. This paper studied the effects of a pore-forming agent addition and preparation temperature on the pore-forming characteristics of ceramsite. The effects of different pore characteristics and external environmental conditions on the water absorption and release properties of the ceramsite were analyzed and the mechanism of water absorption and the release of ceramsite were studied. The study also analyzed the effect of internal metals' curing of ceramsite. A preliminary assessment of the environmental risk of ceramic pellets in the application process was also conducted. This has a very important significance in the construction of sponge cities.

## 2. Materials and Methods

### 2.1. Raw Materials

The raw materials used in the experiment are lake dredging sediment and cyanobacterial slurry, both from East Lake in Wuhan. The dry chemical composition is listed in Table 1.

**Table 1.** Main components and contents of raw materials (wt%).

| Raw materials | $SiO_2$ | $Al_2O_3$ | $Fe_2O_3$ | MgO | CaO | $K_2O$ | Ni | Mn | Cr | Pb |
|---|---|---|---|---|---|---|---|---|---|---|
| Dredging sediment | 57.95 | 16.66 | 9.00 | 2.37 | 6.66 | 3.61 | 0.024 | 0.224 | 0.051 | 0.017 |
| Cyanobacterial slurry | 18.93 | 35.89 | 10.54 | 1.77 | 10.92 | 5.56 | 0.059 | 0.384 | 0.101 | 0.040 |

### 2.2. Ceramsite Preparation Method and Experimental Equipment

The preparation of ceramsite consists of five steps: crushing and sieving, aging, granulation, drying, and low–high temperature pyrolysis. After the lake sediment and cyanobacteria pulp were retrieved, they were dried naturally at room temperature, crushed and sieved, and the 0.17 mm sieved material was kept as a reserve. The corresponding proportion of sediment and cyanobacteria powder was weighed in a mixer, the appropriate amount of ultra-pure water for mixing was added, and then the sediment pellets were aged for 24 h. The aged sediment pellets were granulated in a granulator (E-25, ShenzhenXinyite Science and technology company, Shenzhen, China) to form raw material with a diameter of 1.5 cm. The drying and low–high temperature pyrolysis steps were implemented through a high-efficiency pyrolysis device for polluted sediment resourcing, mainly composed of three parts: an in line drying module, a two-stage controlled temperature pyrolysis module, and a flue gas treatment module. The chain drying module mainly comprised a mesh belt kiln, transmission equipment, and temperature control equipment. After the raw material entered the drying module, the volatile organic matter and moisture could be removed by controlling the temperature and the residence time. The dried raw material entered the low–high temperature pyrolysis module through the connecting device. The purpose of low temperature pyrolysis was to decompose the organic matter in the raw material to produce gas and to control the organic matter content entering the high temperature pyrolysis section. High temperature pyrolysis enhanced the stability of the pore structure of ceramsite by organic matter and decomposable salts (such as carbonates). The cyanobacteria-sediment-based ceramsite can be obtained after the low–high temperature pyrolysis stage. The low–high temperature pyrolysis time was determined by the rotating speed of the rotary in the middle of the module. The gas generated by the pyrolysis process was collected by the flue gas treatment module at the discharge port and then re-entered into the rolling kiln to reduce the loss of energy and emission of polluted gas. The process flow diagram is listed in Figure 1.

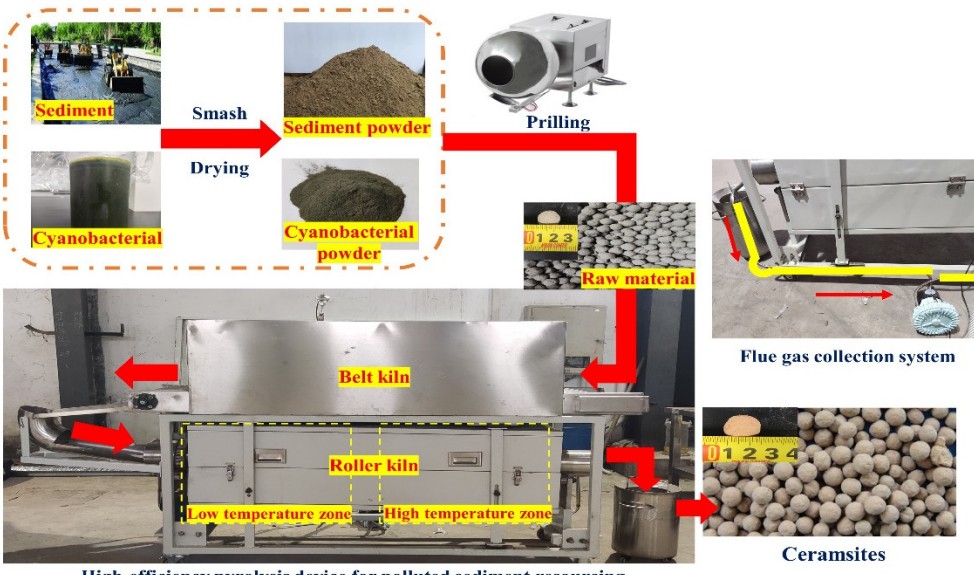

**Figure 1.** Ceramsite production process flow chart.

*2.3. Research Content*

This experiment mainly explored the effects of different cyanobacteria-to-sediment ratios, low temperature pyrolysis temperature, and high temperature pyrolysis temperature on the pore distribution and pore-forming characteristics of ceramsite. The experiment investigated the material's water absorption and release characteristics and the environmental risks in the application process. The specific experimental setting is given in Tables 2 and 3.

**Table 2.** Set table of experimental protocol for ceramsite production.

| Experiments | Cyanobacteria-to-Sediment Ratios | Low Temperature Pyrolysis (°C) | High Temperature Pyrolysis (°C) |
|---|---|---|---|
| Set I | 0:10<br>1:9<br>2:8<br>3:7<br>4:6<br>5:5 | 500 | 1050 |
| Set II | 3:7 | 400<br>450<br>500<br>550<br>600<br>650 | 1050 |
| Set III | 3:7 | 500 | 900<br>950<br>1000<br>1050<br>1100<br>1150 |

**Table 3.** Set table of experimental scheme of ceramsite performance.

| Experiments | Materials | Mass (g) | Ambient Temperature (°C) | Water Ambient Temperature (°C) | pH |
|---|---|---|---|---|---|
| Set I | Ceramsite | 2 | 40 | - | - |
| | Soil | 2 | 40 | - | - |
| | Red brick | 2 | 40 | - | - |
| | Dinas | 2 | 40 | - | - |
| Set II | Ceramsite | 1 | 45 | - | 3 |
| | | 1 | 45 | - | 5 |
| | | 1 | 45 | - | 7 |
| | | 1 | 45 | - | 9 |
| | | 1 | 45 | - | 11 |
| Set III | Ceramsite | 1 | 5 | - | 9 |
| | | 1 | 15 | - | 9 |
| | | 1 | 25 | - | 9 |
| | | 1 | 35 | - | 9 |
| | | 1 | 45 | - | 9 |
| Set IV | Ceramsite | 2 | 40 | - | 9 |
| Set V | Ceramsite | 1 | 25 | - | 3 |
| | | 1 | 25 | - | 5 |
| | | 1 | 25 | - | 7 |
| | | 1 | 25 | - | 9 |
| | | 1 | 25 | - | 11 |
| Set VI | Ceramsite | 1 | 5 | - | 9 |
| | | 1 | 15 | - | 9 |
| | | 1 | 25 | - | 9 |
| | | 1 | 35 | - | 9 |
| | | 1 | 45 | - | 9 |

*2.4. Analytical Methods*

The chemical composition of the raw materials used in the experiments was determined by X-ray fluorescence (Thermo Scientific ARL Perform'X: PANalytical) spectrometry [23]. The physical phase composition of the ceramsite was analyzed by X-ray diffractometers (D8 ADVANCE: Brock Co., Karlsruhe, Germany) [24]. The microstructure of the materials was characterized by scanning electron microscopy. A comprehensive thermal analyzer analyzed the mass change law of sediment and cyanobacteria when subjected to heat to obtain the pore-forming conditions of the ceramsite. Then, the pore distribution of the ceramsite was analyzed by automatic specific surface area and an air void analyzer.

The stacking density and air void were used as the evaluation indexes of the performance of ceramsite, as shown in Equations (1) and (2).

$$\rho = \frac{m}{V_1} \tag{1}$$

$$P = 1 - \frac{m(V_2 - V_3)}{mV_1} \times 100\% \tag{2}$$

In the formula, $\rho$ is the stacking density, $Kg \cdot m^{-3}$; p is the air void, %; m is the mass of ceramsite, g; $V_1$ is the apparent volume of the ceramsite after pouring into the measuring cylinder, mL; $V_2$ is a volume of ceramsite after adding deionized water, mL; $V_3$ is the volume of deionized water in the dosing tube, mL.

*2.5. Water Absorption and Release Effect Test of Ceramsite*

The water release efficiency of ceramsite was determined as follows in four kinds of materials: ceramsite, soil, dinas, and red brick. After crushing to 0.17 mm, 2 g of sieved

material was weighed in the test tube and filled with ultrapure water. After the water release material reached saturation, it was placed in an oven at 40 °C to simulate natural conditions for water release research [25]. The calculation formula is shown in Equation (3).

$$W_r = 1 - \frac{m_1 - m_2}{m_1} \times 100\% \tag{3}$$

In the formula, $W_r$ is the real-time moisture content, %; $m_1$ is the total mass of ceramsite after absorbing water, g; $m_2$ is the total mass after real-time moisture release, g.

The solutions with a pH of 3, 5, 7, 9, and 11 were added to 1 g of ceramsite. After the ceramsite was saturated, the water release experiment was carried out in a constant temperature oven at 45 °C to explore the water release results of ceramsite under different pH conditions. Then, 1 g of water-storing clay was weighed, added with a solution of pH = 9 for water absorption, and placed in a constant temperature oven at 5, 15, 25, 35, and 45 °C for water release experiments to explore the water release results of ceramsite at different temperatures. The solubility of hydrochloric acid and alkali solubility was calculated as shown in (4) and (5).

$$W_h = \frac{1 - m_3}{1} \times 100\% \tag{4}$$

$$W_a = \frac{1 - m_4}{1} \times 100\% \tag{5}$$

In the formula, $W_h$ is the solubility of hydrochloric acid, %; $W_a$ is the alkali solubility, %; $m_3$ is the total mass of ceramsite after soaking in an acidic solution, g; $m_4$ is the total mass of ceramsite after soaking in an alkaline solution, g.

### 2.6. Environmental Risk Test of Ceramsite

The metals' retention rate was determined as follows: after the ceramsite was crushed, 1 g of the sieved material after 0.17 mm sieving was weighed into a conical flask, the conical flask was fixed vertically on a constant temperature shaker, and 100 mL of water was added to adjust the pH and temperature. The vibration frequency was set to 100 times/min and the shaking time was 8 h. After the oscillation was completed, it was allowed to stand for 16 h. The leaching solution was filtered through a 0.45 μm filter membrane and collected to determine the metals' content [26]. For the concentration of Cr, Fe, Mn, Ni, and Pb metals in the material, respectively, they were determined in accordance with the method specified in HJ557-2010.

### 3. Results and Discussion

### 3.1. Effect of Raw Material Ratio on Pore-Forming Characteristics of Ceramsite

The effects of different cyanobacteria-to-sediment ratios on pore-forming characteristics of ceramsite are listed in Figure 2.

The microstructural characterization of the ceramsite was performed by adding different cyanobacteria-to-sediment ratios. The results are shown in Figure 2a–f. As the cyanobacteria-to-sediment ratio increased, the internal pore structure of the ceramsite became more and more abundant. It can be seen from Figure 2a, b that when the cyanobacteria-to-sediment ratios were low, the gas pressure produced by the pyrolysis of organic matter was low, which was not enough to break through the surface tension of the ceramsite [16,27], and the pore structure was mainly composed of micropores and mesopores. From Figure 2 c–f it can be seen that with the increase in cyanobacteria-to-sediment ratios, the pyrolysis of organic matter in raw materials produced a large amount of gas, which increased the internal pressure of ceramsite. At high temperatures, the gas diffused outward, the pore structure distribution of ceramsite began to change, and the proportion of connected pores and macropores increased.

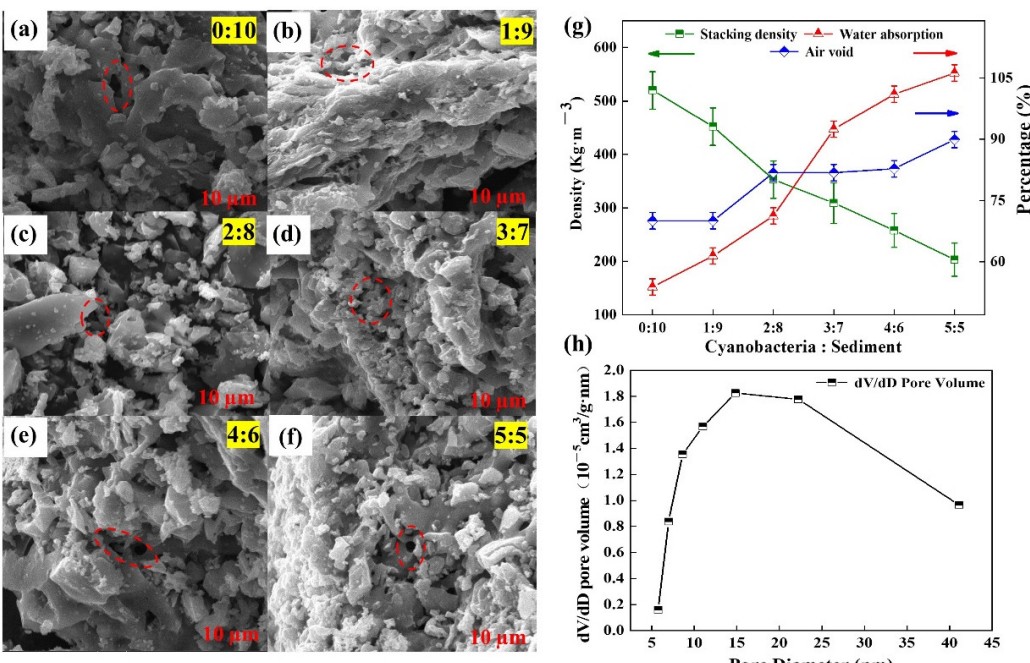

**Figure 2.** Effect of different cyanobacteria-to-sediment ratios on properties of ceramsite. (**a–f**) are electron microscope structure diagrams of water storage materials under different cyanobacteria-to-sediment ratios, (**g**) is stacking density and air void, and (**h**) is pore size distribution.

It can be seen from Figure 2g that with the increase in cyanobacteria-to-sediment ratios, the air void of the ceramsite increased gradually and the stacking density decreased gradually. When the cyanobacteria-to-sediment ratio was between 0:10 and 5:5, the stacking density of the ceramsite decreased by 317 Kg·m$^{-3}$, while the air void increased by 29.9%. When the organic matter content in the raw material increased, the pyrolysis gas production of organic matter increased, which increased the internal air void of the material [28]; the increase in the air void of the multi-ceramsite in the same volume led to the reduction of the internal skeleton support of the material and then the stacking density of the ceramsite decreased [29]. When the cyanobacteria-sediment ratio was more than 5:5, the ceramsite began to break and was difficult to form; therefore, the cyanobacteria-to-sediment ratio ended at 5:5 in the experiment and the cyanobacteria-to-sediment ratio was 3:7 in the subsequent stage.

The BET analysis shows that the specific surface area of ceramsite was 0.65 m$^2$·g$^{-1}$, the micropore volume was 4.14 × 10$^{-4}$ cm$^3$·g$^{-1}$, and the average pore diameter was 31.58 nm (Table S1). The N$_2$ adsorption–desorption isotherm curve of ceramsite conformed to the type III adsorption–desorption curve defined by the International Union of Pure and Applied Chemistry (IUPAC) (Figure S1), indicating that there were large pores inside the material. From Figure 2h, it can be seen that the internal pore size of the ceramsite was between 5–45 nm, indicating that the internal mesopores of the material accounted for the largest proportion and the most probable pore size was 14.82 nm. Because of the small pore volume of ceramsite, the proportion of micropores in the ceramsite was relatively tiny [30].

The above images were processed in two dimensions using Image Pro Plus 6.0 (Media Cybernetics) and the results are in Figure S2. Table S2 is the proportion of pores per unit area of ceramsite prepared by different cyanobacteria-to-sediment ratios. It can be seen from Table S2 that the cyanobacteria-to-sediment ratio was proportional to the internal air void of the ceramsite, which is consistent with the previous experimental results.

*3.2. Effect of Low Temperature Pyrolysis Temperature on Pore-Forming Characteristics of Ceramsite*

It is necessary to determine the temperature limit of the low temperature pyrolysis reaction to ensure that the organic matter in the ceramsite raw material can be fully py-

rolyzed and pore-forming. Thermogravimetric analysis was carried out on sediment and cyanobacterial algal powder. The results are shown in Figures 3 and 4.

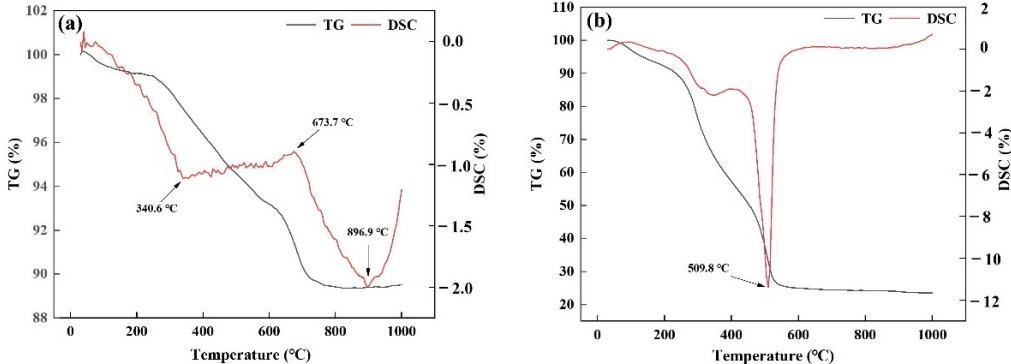

**Figure 3.** TG curves of sediment (**a**) and cyanobacterial powder (**b**).

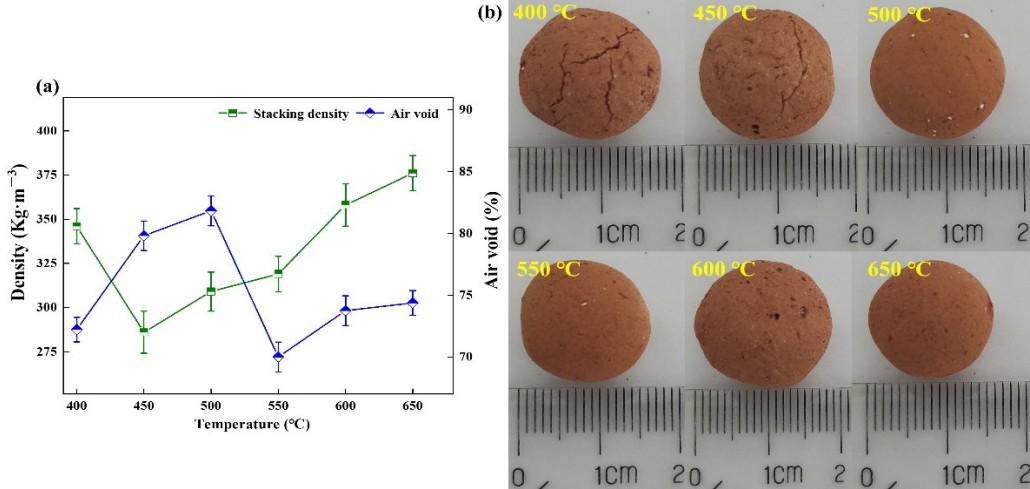

**Figure 4.** Effect of low temperature pyrolysis temperature on the pore structure of ceramsite ((**a**) is stacking density and air void, (**b**) is ceramsite prepared by low temperature pyrolysis).

From Figure 3a,b, it can be seen that the evaporation of water and volatiles inside the sediment made the material decay in mass at 0–340.6 °C. As the temperature increased to 673.7 °C, the organic matter inside the material started to pyrolyze gradually. The pyrolysis gas production made the material mass decrease. When the temperature increased to 896.9 °C, the material became further exothermic but the mass remained unchanged. This indicated that the material started to crystallize and solidify internally. The weight of cyanobacteria powder decreased more at 509.8 °C. For fully pyrolyzing the organic matter in sediment and cyanobacteria into pores, the low temperature pyrolysis temperature range was determined by the future thermogravimetric material analysis.

The degree of reaction in the low temperature pyrolysis section of organic matter on the pore-forming characteristics of ceramsite was further studied. The low temperature pyrolysis temperatures were 400 °C, 450 °C, 500 °C, 550 °C, 600 °C, and 650 °C, respectively, and underwent pyrolysis at 1050 °C. The results of the formation of ceramsite are shown in Figure 4. It can be seen from Figure 4a that with the change of low temperature pyrolysis temperature, the stacking density of ceramsite showed a trend of decreasing first and then increasing, while the air void increased first and then decreased. When the low temperature pyrolysis temperature was 500 °C, the air void reached the highest, which was 81.82%.

The pore structure of the material was related to the reaction degree of organic matter in the low temperature pyrolysis stage. It can be seen from Figure 3 that the organic

matter began to decompose at 340.6 °C and the organic matter reacted utterly when the temperature reached 673.7 °C.

As a pore-forming agent, organic matter can form a rich pore structure by pyrolysis, which increases the air void of ceramsite. When the pyrolysis temperature was less than 500 °C, the reaction of the organic matter was not complete and the remaining organic matter was brought into the high temperature pyrolysis section, which made the internal gas production increase sharply in a short time and cracks appeared on the surface of the ceramsite (Figure 4b) and the internal air void of the material decreased.

When the pyrolysis temperature was higher than 500 °C, the organic matter reacted completely, the gas production in the high temperature pyrolysis section was small, and it was difficult to support the original pore structure in the high temperature molten state [31], resulting in the collapse and reduced air void inside the material.

When the low temperature pyrolysis temperature was 500 °C, the gas generated by the pyrolysis of organic matter brought into the high temperature section could effectively support the internal pore structure of the material so that the surface structure of the ceramsite was complete, the internal pores were abundant, and the air void increased [3].

### 3.3. Effect of High Temperature Pyrolysis Temperature on Pore-Forming Characteristics of Ceramsite

The high temperature pyrolysis treatment was required to ensure the strength of ceramsite. The material after low temperature pyrolysis at 500 °C entered the high temperature section, and different high temperature pyrolysis reaction temperatures were controlled to generate ceramsite, as shown in Figure 5.

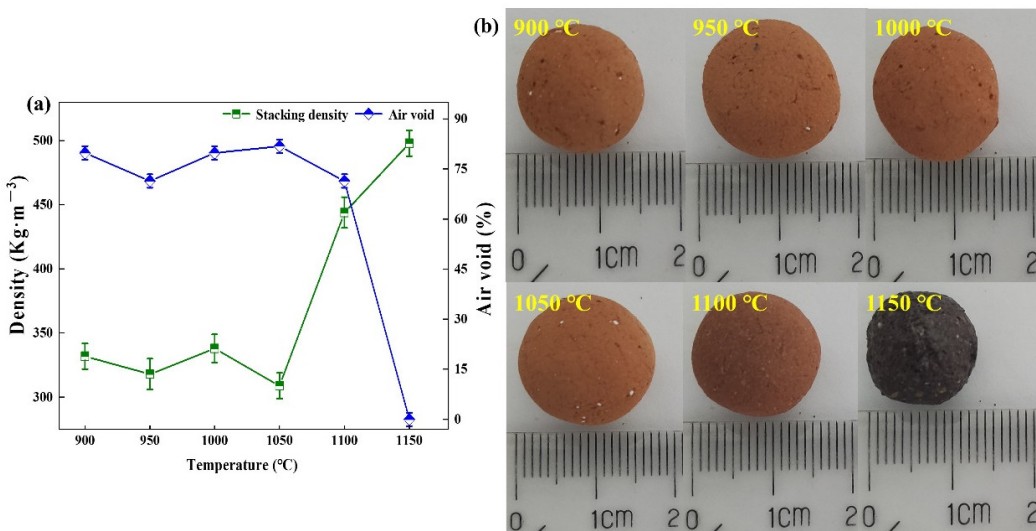

**Figure 5.** Effect of high temperature pyrolysis temperature on ceramsite ((**a**) is stacking density and air void, (**b**) is a physical picture).

It can be seen from Figure 5a that when the high temperature pyrolysis temperature was 950 °C, the air void of the ceramsite decreased and then, with the increase of the high temperature pyrolysis temperature, the air void of the ceramsite increased first and then decreased. At 1050 °C, the air void reached the highest at 81.82%. The reason is that as the temperature rose to 900 °C, the easily decomposed salts inside the ceramsite began to produce gas and the internal pore structure was further enriched. When the temperature reached 950 °C, the material's internal structure collapsed due to softening and the air void decreased [32,33]. With the increase in temperature, the material interior began to crystallize and produced the corresponding skeleton structure, so inside the ceramsite many pore structures were formed [34]. When the high temperature pyrolysis temperature rose to 1150 °C, the surface of the material began to be smooth, the color of the ceramsite

changed from brown to purple–black (Figure 5b), the volume decreased, and the air void was almost zero.

The internal crystallization behavior of ceramsite was studied by XRD. As shown in Figure 6, at 1000 °C, the main crystalline phases in the ceramsite were $CaCO_3$, $CaMg(CO_3)_2$, $Al_2O_3$, and $SiO_2$. When the temperature rose to 1050 °C, the crystal phases of $CaCO_3$, $Al_2O_3$, and $SiO_2$ in the ceramsite disappeared and were replaced by anorthite ($CaAl_2Si_2O_8$) and spinel ($MgAl_2O_4$), because when the temperature reached 1050 °C, the internal $CaCO_3$ was completely decomposed, $CaMg(CO_3)_2$ was converted into $MgAl_2O_4$ and $CaAl_2Si_2O_8$, and the crystal structure of $MgAl_2O_4$ and $CaAl_2Si_2O_8$ made the ceramsite structure stable.

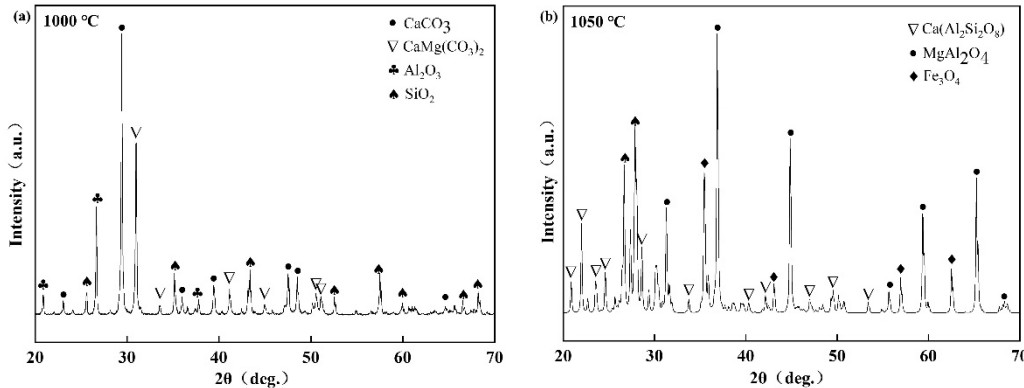

**Figure 6.** XRD patterns of ceramsite at 1000 °C (**a**) and 1050 °C (**b**).

### 3.4. Study on Water Absorption and Release of Ceramsite

Ceramsite has a rich pore structure. Studying its water absorption and release performance is necessary to ensure its application in sponge city construction. Four kinds of materials with different pore structures—ceramsite, soil, red brick, and dinas—were selected to study the water release capacity of each material at a constant room temperature of 40 °C after total water absorption.

It can be seen from Figure 7 that the saturated water absorptions of ceramsite, soil, red brick, and dinas were 92.45%, 84.45%, 67.05%, and 33.25%, respectively, when the mass was 2 g. Ceramsite was higher than other materials in the water absorption effect. The water release experiment showed that the water retention time of ceramsite was 43 h, 59 h, and 74 h more than that of soil, red brick, and dinas at a room temperature of 40 °C, respectively, which showed an obvious water retention advantage. The reason is that the internal air void of ceramsite is relatively rich in other materials such as soil, red brick, and dinas. Therefore, ceramsite has more water absorption and release pore structure for different pore materials of the same quality, resulting in better water absorption and release performance than other materials.

The water storage materials were analyzed under different pH conditions. The results are listed in Figure 8. It can be seen from Figure 8 that, when the pH was 3, the water absorption rate of ceramsite reached the highest at 95.2%, compared with other conditions. After 30 h of water release, the water release rate of ceramsite under acidic conditions increased significantly. The reason is that under acidic conditions, part of the pore walls of the internal pore structure of ceramsite was destroyed and the salt dissolution rate was 2.2%. Due to the destruction of the pore wall structure, the internal pore structure of the ceramsite was connected, the water absorption capacity was enhanced, and the water retention capacity was reduced. At pH 11, the alkaline solubility of ceramsite was 1.3%. It shows that the damage strength of the pore structure of ceramsite under alkaline conditions was weaker than that under acidic conditions. In the later water release experiment, the water release rate was lower than under acidic conditions. Due to the electronegativity of inorganic materials, the internal charge of ceramsite was changed under acidic conditions, affecting the water release results.

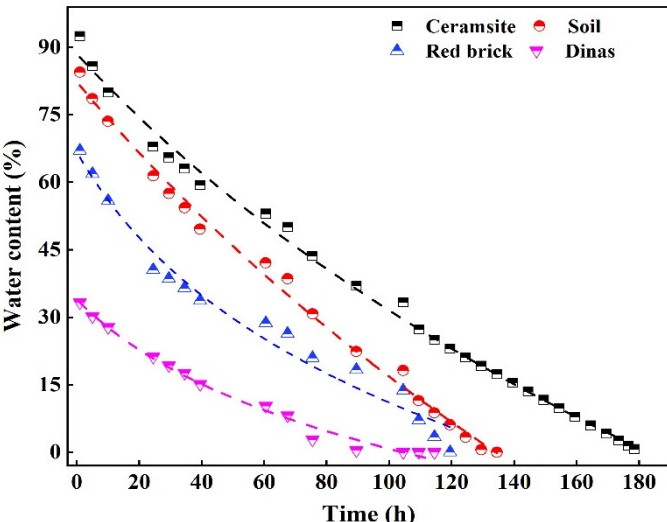

**Figure 7.** Water absorption and release capacity of different pore materials.

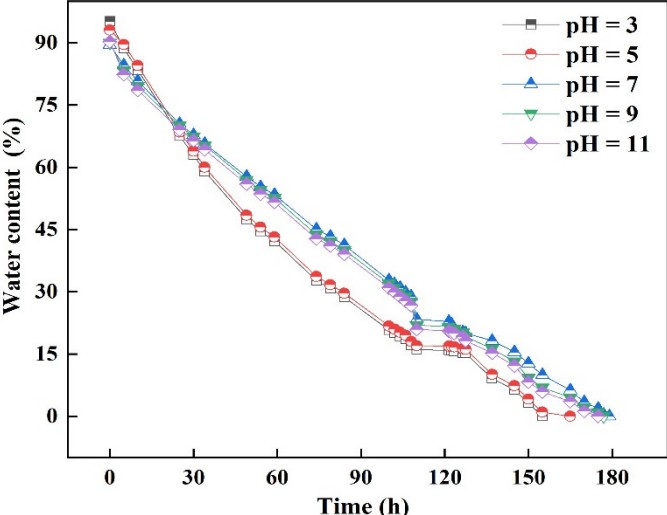

**Figure 8.** Water release capacity of water storage materials under different pH conditions.

The water storage materials were analyzed under different temperature conditions. The results are shown in Figure 9. It can be seen from Figure 9 that, with the increase in temperature, the water release time of ceramsite gradually shortened. Compared with the release temperature of 5 °C, the release time at 45 °C was reduced by 116 h. Under a high temperature environment, the average kinetic energy of water molecules inside the ceramsite increased and the irregular movement of molecules was intense, which shortened the water release time inside the ceramsite [35].

The effect of repeated water absorption and the release of ceramsite was further studied. It can be seen from Figure 10 that the ceramsite can still maintain good water absorption and release performance after five cycles of absorption and release. The water absorption rate of the ceramsite was stable at 92.45 ± 2% and the water release time was stable at 178.5 ± 5 h. The BET results show that the ceramsite contained abundant macroporous and mesoporous structures. Because of macropores inside the material, the water absorption performance of ceramsite was good. Due to the effect of mesopores, the ceramsite had a long water release time, a stable water release structure, and was not easy to be destroyed, showing an excellent cyclic water absorption and release effect [36].

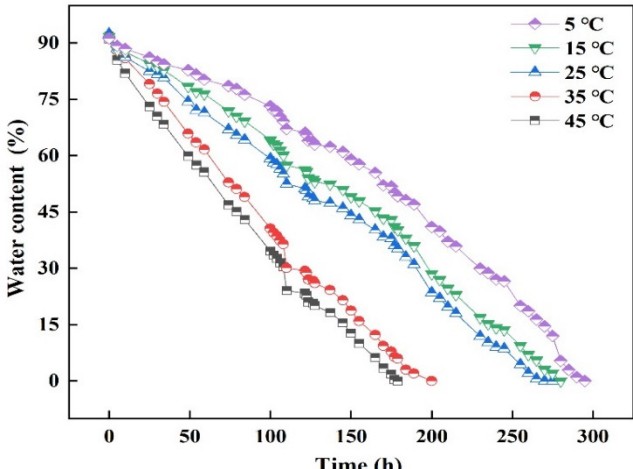

**Figure 9.** Water release capacity of water storage materials at different temperatures.

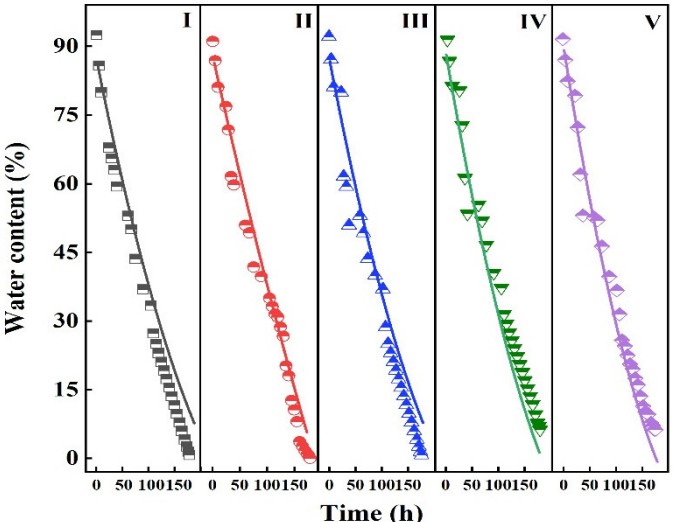

**Figure 10.** Effect of cyclic water absorption and release of ceramsite.

*3.5. Risk Assessment of Metals' Dissolution in Ceramsite*

The risk of metals' dissolution should be fully considered in a ceramsite application. The application of ceramsite in acidic–alkaline soil and the dissolution of metals at different temperatures were investigated. The results are shown in Figures 11 and 12.

It can be seen from Figure 11a that Cr in cyanobacteria-sediment-based ceramsite had a higher dissolution at pH 5 and 9. Cr is prone to hydrolysis, therefore increases dissolution under acidic or alkaline conditions. The hydrolysis reaction is reversible, therefore the dissolution is reduced under the conditions of excessive acid or excessive alkali. It can be seen from Figure 11b that Fe in ceramsite increased first and then decreased with the increase of pH value. When the pH value was 7, the dissolution of Fe reached $4.83 \times 10^{-2}$ mg·g$^{-1}$. Fe is an important part of ceramsite sintering in the firing process, mainly in skeleton construction. Under acidic and alkaline conditions, the structure is not easily destroyed and the dissolution amount is small. It can be seen from Figure 11c that the Mn in the ceramsite showed a gradual decrease with the increase of the pH value. When the pH value was 7, the dissolution of Mn slightly increased. Mn exhibited an ion-exchangeable state under acidic conditions. When the pH value gradually increased, Mn and OH$^-$ in the solution formed a white precipitate, resulting in less Mn dissolution. It can be seen from Figure 11d that Ni in the ceramsite decreased first and then increased with the increase of the pH value. The reason may be that under stable acid conditions,

the crystal form of Ni was destroyed, resulting in increased dissolution. Under alkaline conditions, the competitiveness of Ni was weaker than that of other substances in the clay, which reduced the dissolution. It can be seen from Figure 11e that the Pb in the ceramsite showed a gradual decrease with the increase of the pH value, because $Pb^{2+}$ combines with $OH^-$ in an alkaline environment to form precipitation and fixes in the pore structure of ceramsite, which made the ceramsite dissolve less under alkaline conditions [37].

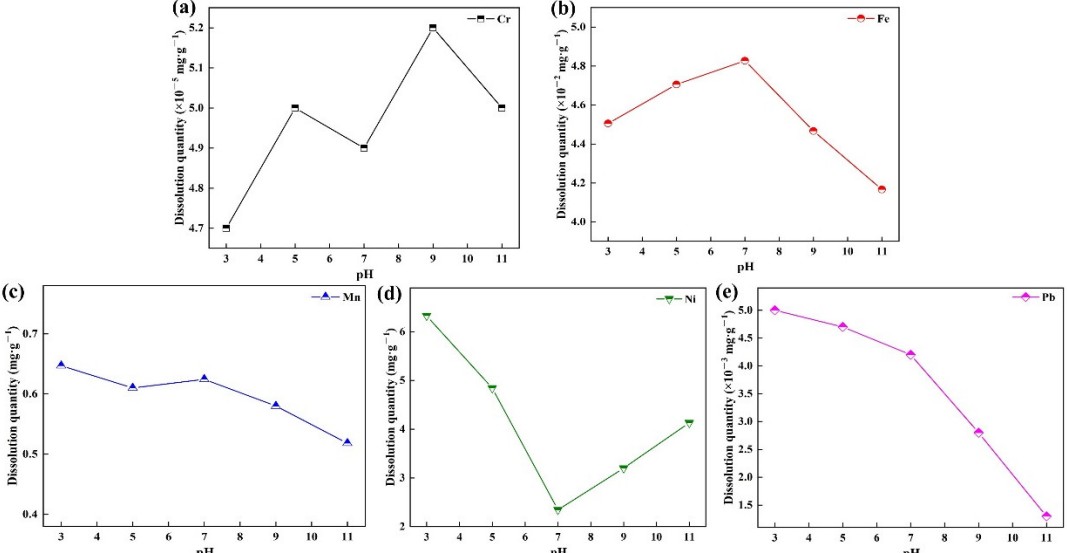

**Figure 11.** Dissolution effect of metals under different pH conditions. (**a**) is the dissolution curve of Cr; (**b**) is the dissolution curve of Fe; (**c**) is the dissolution curve of Mn; (**d**) is the dissolution curve of Ni; (**e**) is the dissolution curve of Pb.

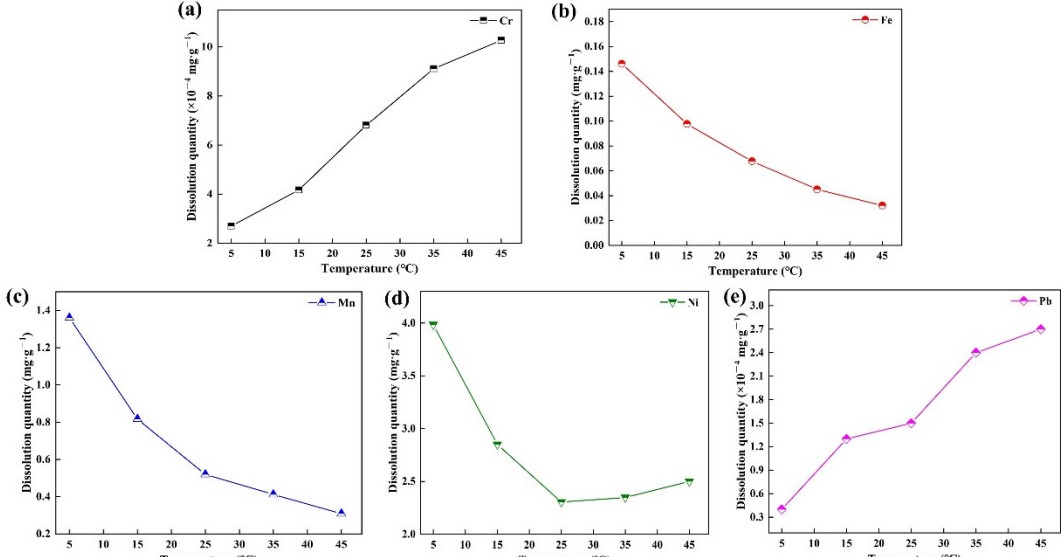

**Figure 12.** Dissolution effect of metals at different temperatures. (**a**) is the dissolution curve of Cr; (**b**) is the dissolution curve of Fe; (**c**) is the dissolution curve of Mn; (**d**) is the dissolution curve of Ni; (**e**) is the dissolution curve of Pb.

It can be seen from Figure 12 that Cr and Pb in cyanobacteria-sediment-based ceramsite showed a gradual increase with the increase in temperature. The dissolution of Fe, Mn, and Ni decreased with the increase in temperature. The increase in temperature made the water-soluble or carbonate-bound metals on the surface of ceramsite dissolve and then the dissolved amount increased. The gradual decrease of Fe, Mn, and Ni was mainly due to the

complexation of Fe, Mn, and Ni with the inorganic ligands on the surface of the ceramsite with the increase in temperature. After the subsequent filtration of the filter membrane, the complex was intercepted on the filter membrane's surface, which made it decrease with the increase of temperature [38].

*3.6. Water Absorption and Release Mechanism of Ceramsite*

Based on the above experiments, the water absorption and the release mechanism of ceramsite can be obtained. This is shown in Figure 13. There are many macroporous structures inside the ceramsite with connecting holes. When water is absorbed, the internally connected pores of the material provide a bridge for the entering water molecules. After water molecules enter the macropores through the interconnected pores, they penetrate the mesopores and micropores through the capillary pores. The water entering the pore structure is combined with the O atoms between the lattices to produce different surface hydroxyl adsorption sites. In addition, the combination of different metals and oxygen atoms in the ceramsite makes the ceramsite appear electronegative [39] and the hydrophilicity of the material is enhanced. Due to the existence of the mesoporous structure in ceramsite, the adsorption sites of surface hydroxyl groups increase and hydrogen bonds are easily formed between surface hydroxyl groups and water molecules. Hydrogen bonds are reversible, which makes ceramsite have better water absorption and release ability [40]. During water release, because of the rich pore space inside the ceramsite, the water evaporates through a longer path than other materials with poor pore structure, so the water release rate is slow [41].

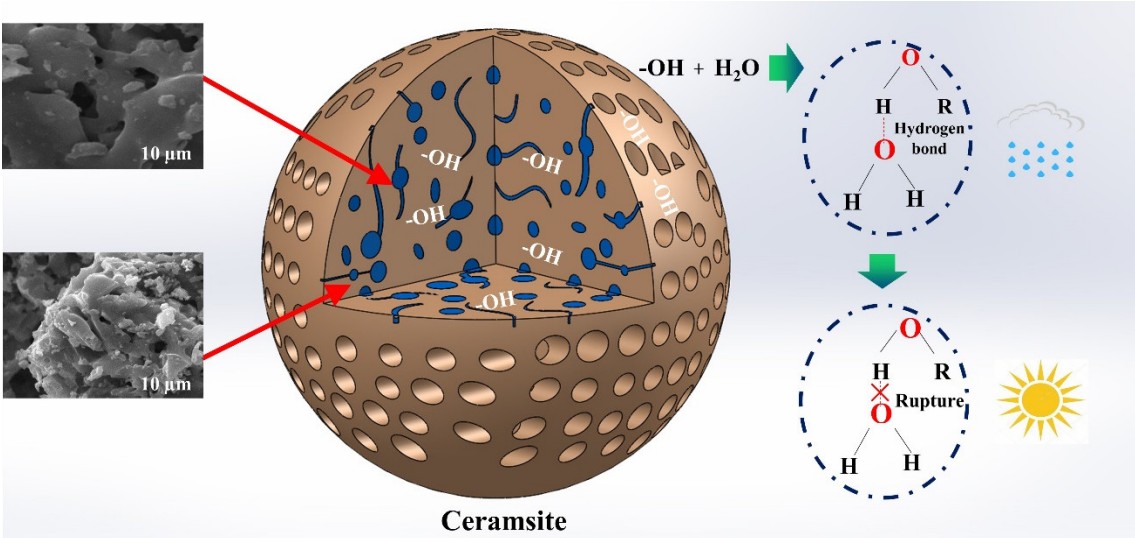

**Figure 13.** Diagram of water absorption and release mechanism of ceramsite.

**4. Conclusions**

In summary, lake sediments and cyanobacteria powder were used as raw materials to make ceramsite, provided a way of resource utilization of lake waste, and alleviated the shortage of mineral resources in China. The main conclusions of this work could be generalized as follows.

- The pore formation of ceramsite is mainly classified into two stages: low temperature organic matter pyrolysis and high temperature decomposable salt pyrolysis. When the low temperature pyrolysis temperature is 500 °C and the high temperature pyrolysis temperature is 1050 °C, the pores of ceramsite are abundant and the proportion of pores per unit area is 80.12%.
- Through BET characterization analysis, it is found that the proportion of macropores is the determinant of the water absorption performance of ceramsite and mesopores are the main reason for the excellent water release effect of ceramsite.

- XRD characterization analysis: $CaAl_2Si_2O_8$ and $MgAl_2O_4$ in the ceramsite is the reason for the material's pore structure stability and the decisive factor for the recycling of the material.
- The water release experiment found that the ceramsite pore structure would be destroyed under acidic and alkaline conditions and the water release time would be reduced.
- Through the analysis of metals' dissolution, it is found that the retention rate of Fe, Ni, Mn, Cr, and Pb is more than 99% under different conditions and the environmental application is safe. It is applied to sponge city construction, highway slope protection, and community landscape.

**Supplementary Materials:** The following supporting information can be downloaded at: https://www.mdpi.com/article/10.3390/pr10112331/s1, Figure S1: Ceramic pellet adsorption and desorption isotherms; Figure S2: Two-dimensional SEM maps with different organic matter contents; Table S1: Analyze data; Table S2: Analyze data.

**Author Contributions:** Conceptualization, Y.H., X.D. and J.P.; validation, Y.H., K.L. and B.L.; resources, Y.H., C.Z. and Z.D.; funding acquisition, Y.H. and W.X.; formal analysis, K.L.; data curation, W.X.; writing—original draft preparation, Y.H. and K.L.; methodology, K.L. All authors have read and agreed to the published version of the manuscript.

**Funding:** This research was funded by Guangdong Water Conservancy Science and Technology Innovation Project (2020-01) and Hubei University of Technology 2020 Doctoral Research Startup Fund Project (BSQD2020045). This research was supported by the National Natural Science Foundation of China (Grant No.: 21906048).

**Data Availability Statement:** Not applicable.

**Conflicts of Interest:** The authors declare no conflict of interest.

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
