# Peer review of "Resource Utilization of Lake Sediment to Prepare “Sponge” Light Aggregate: Pore Structure and Water Retention Mechanism Study"

_processes, doi:10.3390/pr10112331_

Round 1

Reviewer 1 Report

The work deals with a very interesting topic, the experimental campaign is appropriate and the experimental procedures are described with sufficient details. As a general content, English should be revised since throughout the paper there are many errors (especially sentences without the verb).

Some minor changes are required before publications

Introduction: Provide a brief explanation of the “sponge city” concept

Section 2.3: It is not clear from the table 1 if the inorganic materials were tested in all the conditions specified in the sets 1, 2 and 3. Actually, the table is not clear since I guess that it reports the conditions of both ceramsites production and performance tests. It would be better to create 2 separate tables.

Section 2.5: Please, specify how metals content was measured. Does the leaching test reproduce the conditions established during exposure of the ceramsite to atmospheric agent? Is it a standard procedure?

In Figures S1, please specify the differences between the 2 plots. Do they refet to adsorption and desorption process?

In table S2, please specify the meaning of Per Area (Obj/total) and how it has been calculated.

Section 3.2: from Fig 3a any weight loss can be observed at 896.9 °C

Section 3.4: Characteristic times of adsorption should be also measured since it is an important parameter to take into account in managing floods in urban areas.

Reviewer 2 Report

This paper presents a comprehensive study of an interesting symbiotic use of two coexisting waste streams: algal bloom biomass and eutrophic lake sediment. 

(1) My main criticism is that there are quite a few points where I think the wrong word has been substituted during translation, so a technically versed editorial review of the English is require, as these currently confuse the reader.  Here are the main ones I noticed during my review.

Line 17 "accumulate" (not "sunk")

Line 21 "compounds or salts" (not "salt")

Line 29  "retention" (not "consolidation")

Line 80 "fly" (not "flying")

Line 130 "in line" (not "chain")

Line 141 "rotary" (not "rolling" kiln)

Line 188 "release" (not "consolidation") but see above

Line 203 delete "electronic", "microstructural" not "microstructure"

Line 444  "retention" or "release" (not "consolidation") 

Line 473  Authors are abbreviated in reference.

(2) Line 117 Table 1.  These are described as "dry" oxide analyses but are they not ash compositions?  If not what was the organic C/volatile content/loss on ignition, as these appear to total 94-83%?

(3) Lines 139, 294 and 339.  What are the "decomposable salts"?  Please suggest likely compounds.  Carbonates? Oxides?

(4) Line 152 Table 1 (should be Table 2).  Any replicate analyses?

(5) Lines 169 and 316.  What is "dinas"?  80 mesh, better to give size in mm.  In line 323  "dinas" is substituted by "sand" in the list of compared materials

(6)  Lines 248-9.  "the sediment has two apparent weight losses at 340.6 C, 673.7 C and 896.9 C".  This needs rewording , as these are three inflection points on the DSC curve, so are thermal features related to mass losses.  Other than water loss around 100 C, I suggest the curves show organic matter decay between 340 and 673 and a further exothermic but constant mass change above 896 (melting).

(7) Line 373.  Suggest "metals" not "heavy metals".

Reviewer 3 Report

I have gone through the submitted article under the title “ Resource utilization of lake sediment to prepare "sponge" light 2 aggregate: pore structure and water retention mechanism study” and recommending it for publication

Author Response

Thank you for your affirmation of this article!

Reviewer 4 Report

This paper characterizes “ceramsite prepared from lake sediment and cyanobacterial slurry”.  The study in itself is interesting, important and publishable.  The manuscript is fairly well written with the exception of the abstract.  The run-on sentences makes the abstract reading confusing.  Please split-up these long sentences to shorter ones.

Abstract Line 19: “It was the research object to study the effects of pyrolysis temperature and organic matter content on the pore-forming characteristics of the materials” – please rewrite this sentence, e.g., The objective of this study is to study the effect of pyrolysis temperature and cyanobacterial sediment on porosity of ceramsite!

There are also other place in the manuscript where you can find run-on sentences.  Please shorten them.

The figures could be larger for better visibility of the details.  The font size used for legends in the graphs next to SEM images are too small and difficulty to read.
